# The Role of the Basophil Activation Test in the Diagnosis of Drug-Induced Anaphylaxis

**DOI:** 10.3390/diagnostics14182036

**Published:** 2024-09-13

**Authors:** Maria Czarnobilska, Małgorzata Bulanda, Ewa Czarnobilska, Wojciech Dyga, Marcel Mazur

**Affiliations:** 1Department of Pathophysiology, Jagiellonian University Medical College, Czysta St. 18, 31-121 Krakow, Poland; maria.czarnobilska@uj.edu.pl; 2Department of Clinical and Environmental Allergology, Jagiellonian University Medical College, Botaniczna St. 3, 31-501 Krakow, Poland; gosia.lesniak@uj.edu.pl (M.B.); ewa.czarnobilska@uj.edu.pl (E.C.); wojciech.dyga@uj.edu.pl (W.D.)

**Keywords:** biomarkers, severe allergic reactions, in vitro test, vaccines

## Abstract

The diagnosis of drug-induced anaphylaxis (DIA) is a serious health problem. The Basophil activation test (BAT) is considered a specific in vitro provocation, and compared to in vivo provocation, it is more convenient, cheaper, and safer for the patient. This study aimed to evaluate the usefulness of the BAT in the diagnosis of DIA. This study included 150 patients referred to a reference allergy clinic with suspected drug allergies. All patients underwent a detailed clinical evaluation supplemented with the BAT. Positive BAT results were obtained in two out of 21 patients who were to receive the COVID-19 vaccine. The sensitivity and specificity of the BAT were 40% and 75% for the COVID-19 vaccine, 67% and 58% for DMG PEG 2000, and 100% and 75% for PEG 4000, respectively. Nine out of 34 patients with suspected antibiotic allergies had positive BAT results with 14 different antibiotics. Positive BAT results were also obtained with NSAIDs in two patients and with local anesthetics in three patients. The confirmation of allergy by the BAT improves the safety profile of the diagnostic work-up as it may defer the need for drug provocation, preventing potential anaphylactic reactions.

## 1. Introduction

According to World Allergy Organization (WAO) guidance 2020, anaphylaxis is “a serious systemic hypersensitivity reaction that is usually rapid in onset and may cause death” [1]. During severe anaphylaxis, potentially life-threatening respiratory and/or circulatory system involvement, and even shock, may occur, with or without cutaneous symptoms. The criteria for diagnosing anaphylaxis include typical skin symptoms with the involvement of at least one or more systems or symptoms of decompensation of the respiratory and/or circulatory system after exposure to a known or probable allergen in the patient [1]. According to the European Academy of Allergy and Clinical Immunology, anaphylaxis is “a life-threatening reaction characterized by acute onset of symptoms involving different organ systems and requiring immediate medical intervention”. The most common causes of anaphylaxis in adults include exposure to drugs, mainly antibiotics (especially beta-lactams) and non-steroidal anti-inflammatory drugs [2].

The assessment of drug-induced anaphylaxis (DIA) relies on a patient’s anamnesis and additional tests [3]. Several systems have been developed to quantify anaphylaxis severity; however, none are perfect, and they lack validation [4]. In the 1980s, Ring and Messmer developed a scale for assessing the severity of anaphylaxis, which is often used in clinical practice. According to this classification, grade II includes signs from at least two organs or systems; in grade III, symptoms of circulatory and/or respiratory failure or shock occur, and grade IV means circulatory or respiratory arrest [5].

The diagnosis of a patient after an anaphylactic reaction requires additional tests. Current data from the literature and clinical practice indicate difficulties in the diagnosis of DIA, caused by a lack of specific, sensitive, and safe tests for the detection of the causative drug [6].

In the diagnosis of DIA, skin tests (STs) are highly specific for some drugs, including beta-lactam antibiotics. However, reagents for STs may not contain all important allergen components. Moreover, in vivo tests such as intradermal skin tests (IDTs) and specific provocation challenges carry the risk of anaphylactic reactions and should be performed in a hospital setting [7].

To reduce the risk of a severe reaction during re-exposure to the drug and to avoid the discontinuation of an important treatment (e.g., antihistamine drugs), an in vitro test should be implemented before potential in vivo testing. The specific IgE (sIgE) assay for beta-lactam antibiotics has low sensitivity and is unavailable for many other drugs [7].

The basophil activation test (BAT) is a flow cytometry-based assay in which the expression of activation markers (such as CD63) on the surface of basophils is measured following stimulation with an allergen [8]. The application of the BAT is considered a specific in vitro provocation. Compared to in vivo provocation, it is more convenient and safer for the patient because it only requires blood collection. It is also less expensive because it does not require hospitalization [8].

There is also an emerging diagnostic tool, known as the mast cell activation test (MAT), that might be useful in exploring differences in effector cell function between basophils and mast cells during allergic reactions [9].

DIA is a difficult diagnostic problem, which allergologists especially noticed during the COVID-19 pandemic. Patients were referred to the allergy clinic not only with a history of DIA but often with drug-related non-allergic adverse reactions, an atopic history, and also just because of their fear of vaccination. Meanwhile, the only absolute contraindication to administering a dose of a COVID-19 vaccine is an anaphylactic reaction to any of its ingredients [10].

The aim of this study was to evaluate the usefulness of the BAT in the diagnosis of DIA in adults whenever it was necessary to choose a safe medication. The research hypothesis was that the BAT can replace the specific challenge test in case of anaphylaxis after vaccination for COVID-19.

## 2. Materials and Methods

This study included 150 patients referred to a reference secondary care public outpatient allergy clinic in Krakow, Poland with suspected drug allergies in 2021–2022. We assumed that patients fall into 3 groups. In each group, we wanted to perform a correlation analysis between clinical symptoms and BAT results. We assumed that both variables are measured at least on an interval scale and met parametric assumptions, so Pearson’s correlation test can be used for analysis. Therefore, based on [11], the sample of 150 patients met the requirements. All patients underwent a detailed clinical evaluation. Inclusion criteria of study participants included adult patients with a history of DIA who agreed to participate in this study, while exclusion criteria included pregnancy and breastfeeding. Patients completed a detailed questionnaire administered by the investigators answering questions about the cause and course of drug-related allergic reactions in the past and accompanying allergic diseases presented in Appendix A: Patient questionnaire. The severity of the anaphylactic reaction was assessed on a Ring and Messmer scale (I–IV).

Based on the survey results, patients were classified into 3 groups:Group 1 with indications for allergy diagnosis before administering the COVID-19 vaccine selected from among DIA patients due to COVID-19 pandemic context (history of anaphylactic reaction to the vaccine, history of hypersensitivity to polyethylene glycol (PEG), and hypersensitivity to many drugs from various chemical groups),Group 2 with indications for the diagnosis of allergies to other drugs,Group 3 with no indications for the diagnosis of drug allergies. This was a group of patients referred with a suspected vaccine allergy—due to adverse reactions after vaccination or an incriminating allergy history.

The BAT was performed using medications selected based on the interview. Of the 150 patients referred to the outpatient clinic, 60 were excluded from DIA at the interview stage and 90 qualified for the BAT. There were no study withdrawals as none of the patients dropped out of the study. The assessment of basophil activation is measured on the base of CD63 antigen expression using the Flow2 CAST Basophil Activation Test (Bühlmann Laboratories AG, Schönenbuch, Switzerland) according to the manufacturer’s protocol. Patients’ blood was collected into ethylenediaminetetraacetic acid (EDTA) tubes, stored at 4 °C, and analyzed within 2 h from the blood collection. A total of 50 µL of whole blood was mixed with 100 µL of stimulation buffer, containing calcium, heparin, and IL-3, and 50 µL of stimulants. Anti-FcɛRI monoclonal antibodies and fMLP oligopeptides were used as specific and nonspecific positive controls, and an unstimulated sample served as the negative control. The samples are stained using a 20 µL mixture of two fluorescently labeled monoclonal antibodies: PE-conjugated anti-CCR3 for basophil selection and FITC-conjugated anti-CD63 for basophil activation status determination. After 15 min incubation at 37 °C, 2 mL of erythrocyte lysing solution were added and the samples were centrifuged, resuspended in 0.3 mL of washing buffer, and analyzed on a flow cytometer (FacsLyric, Becton, Dickinson and Company, Franklin Lakes, NJ, USA). Details of flow cytometry analysis and the basophil gating strategy (CCR3+/SSC-low cells) are presented in Figure 1.

The test was considered positive when CD63 expression was >5% and the stimulation index (the ratio of the percentage of CD63-expressing cells with drug exposure/percentage of CD63-expressing cells with wash buffer) was >2. SPTs and IDT of potential culprit drugs were performed in patients who consented to undergo those tests with histamine solution as the positive control and NaCl 0.9% as a negative control (all patients were approached for the tests, five consented for IDT).

Patients tested with COVID-19 vaccines were later vaccinated—in the case of negative BAT results, with the vaccine used for testing, and in the case of positive results, with an alternative vaccine without PEG. We have considered vaccination to be equivalent to a drug provocation challenge test.

This study was approved by the Jagiellonian University Bioethical Committee, and patients provided written informed consent to participate in this study.

### Statistical Analysis

We calculated sensitivity and specificity using clinical history as the gold standard to evaluate BAT performance. The data were compared using various tests such as the unpaired *t*-test, the Mann–Whitney test, analysis of variance (ANOVA), or the Kruskal–Wallis test. Correlations were assessed using Spearman’s or Pearson’s correlations. A result was deemed significant when the *p*-value was less than 0.05. All statistical analyses were conducted using the Julia programming language, version 1.9.4, and the following packages: HypothesisTests.jl, Statistics.jl, DataFrames.jl, and Pluto.jl.

## 3. Results

A total of 150 people completed the survey.

Group 1, with indications for allergy diagnostics before vaccination against COVID-19, included 21 patients with a history of anaphylaxis after the administered dose of the vaccine and hypersensitivity to PEG or many drugs from various chemical groups.

Clinical symptoms occurring after vaccination against COVID-19 in patients from this group are presented in Figure 2.

Positive BAT results were obtained in two people (Table 1).

### 3.1. Patient 1

A 40-year-old patient with no history of chronic diseases or allergies was admitted to the allergology department in order to qualify for vaccination with the third dose of the COVID-19 vaccine. On 4 June 2021, the patient was vaccinated with the first dose of the vaccine with good tolerance. On 9 July 2021, the second dose of the vaccine was administered, after which the patient developed an anaphylactic reaction (Ring and Messner III).

The patient underwent STs with the Comirnaty (Pfizer Inc., New York, NY, USA) vaccine and obtained a positive IDT result for the 1:100 concentration. Due to the positive IDT, in order to confirm an allergy to one of the ingredients of the vaccine, a BAT was performed with positive results: percentage of activated basophils 12.9%, SI = 25.8 for the vaccine; percentage of activated basophils 11.2%, SI = 22.4 for DMG-PEG 2000; and percentage of activated basophils = 9.6%, 19.2 for PEG 4000 (Figure 3B).

### 3.2. Patient 2

Patient 2 was a 24-year-old patient with a history of anaphylaxis (Ring Messner III) 10 min after vaccination with one dose of the Comirnaty (Pfizer Inc., New York, NY, USA) vaccine. The patient had negative STs with the vaccine; negative BAT results for the vaccine (4.8% of activated basophils, SI = 48); and positive BAT results for DMG-PEG 2000 (12.2% of activated basophils, SI = 122) and PEG 4000 (14.7% of activated basophils, SI = 147; Figure 4B).

The Pearson correlation coefficient test (r) of clinical symptoms determined by the RM scale showed:with the percentage of activated basophils for DMG-PEG 2000, there is a statistically significant positive correlation, r = 0.488801; *p* = 0.0395,with SI for DMG-PEG 2000, there is a statistically significant positive correlation, r = 0.519732; *p* = 0.0271,with the percentage of activated basophils for PEG 4000, there is a statistically significant positive correlation, r = 0.636658; *p* = 0.0478,with SI for PEG 4000, there is a statistically significant positive correlation, r = 0.729909; *p* = 0.0166,with the percentage of activated basophils for the vaccine, there is a statistically significant positive correlation, r = 0.766223; *p* = 0.0160,with SI for the vaccine, there is a significant positive correlation r = 0.632157; *p* = 0.0678.

Considering clinical symptoms as the gold standard, the sensitivity and specificity of the BAT were, respectively, 40% and 75% for the vaccine, 67% and 58% for DMG-PEG 2000, and 100% and 75% for PEG 4000.

### 3.3. BAT in the Diagnosis of DIA with Other Drugs

Group 2 included 69 patients. A total of 34 patients underwent the BAT with antibiotics, 14 patients were tested with NSAIDs, seven were tested with local anesthetics, seven with general anesthetic, and seven with radiocontrast media selected on the basis of the medical interview.

Clinical symptoms occurring in patients from group 2 during DIA are presented in Figure 5.

Nine patients had positive BAT results with 14 different antibiotics. In patients with a history of anaphylaxis after the administration of antibiotics, the BAT was performed with the following antibiotics: penicillin, ampicillin, amoxicillin, amoxicillin and clavulanic acid, cefuroxime, ciprofloxacin, clindamycin, and clarithromycin. Positive BAT results were obtained with NSAIDs in two patients and with local anesthetics in three patients. Positive results are presented in Table 2.

A statistically significant positive correlation was obtained for clinical symptoms assessed on the R&M scale and the percentage of activated basophils: r = 0.754018; *p* = 0.0073 and SI: r = 0.678758, *p* = 0.0217 for cefuroxime.

Taking clinical symptoms as the gold standard, the sensitivity and specificity of BAT were, respectively, 100% and 100% for amoxicillin, 60% and 86% for amoxicillin with clavulanic acid, 80% and 67% for cefuroxime, 100% and 100% for ampicillin, and 100% and 55% for clarithromycin.

None of the BAT results in seven patients who were administered radiocontrast media were positive.

Group 3 included 60 patients who, based on the questionnaire and a detailed interview, were not subjected to any allergy diagnosis (Figure 6).

## 4. Discussion

### 4.1. Clinical Characteristics of the Study Population

In our study, the female-to-male ratio was 75% to 25%. This analysis corresponds to the results collected by the European Anaphylaxis Registry in 2023, where the authors indicate that the proportion of female patients with DIA is higher than in other anaphylactic reactions (65.34%), while DIA in males was less reported [12].

### 4.2. Symptom Profile of DIA

The symptoms experienced by most of our patients during DIA include skin symptoms, such as rash, urticaria, and edema. More than half of patients during DIA also reported respiratory symptoms, such as dyspnea, and cardiovascular symptoms, such as a drop in blood pressure or loss of consciousness. Only a few patients had nausea and vomiting. In the European Anaphylaxis Registry, during DIA, patients reported skin and mucous membrane (84.02%), respiratory (71.63%), cardiovascular 68.93%), and gastrointestinal system symptoms (30.25%) [12].

In 2023, Greek authors prospectively assessed 13 patients with a history of 54 drug reactions, performing routine diagnostics supplemented with BATs. The sensitivity of the BAT to drugs obtained by them in clinical practice was 97.6%, and the specificity was 96% for drug allergies [13]. In our study, we assessed the use of the BAT in patients with a history of DIA, taking into account the division into different drug groups. Like the Greek authors, we considered the interview to be the most important point in allergology diagnosis [13].

### 4.3. The Role of the BAT in the Qualification for Vaccination against COVID-19

This study was conducted during the COVID-19 pandemic, which gave it a specific context and allowed for the analysis of a group of patients diagnosed before vaccination. The need to vaccinate the population with new COVID-19 vaccines resulted in many patients being referred to a specialist allergy clinic in order to qualify for vaccination. Vaccination against COVID-19 was seen as an opportunity to prevent serious health consequences and potentially even death. Therefore, it was particularly important to exclude possible life-threatening contraindications, which undoubtedly include the risk of anaphylaxis [14].

According to the guidelines, the only absolute contraindication to the administration of the next dose of a vaccine is an anaphylactic reaction to any of its ingredients [15,16]. According to the document Recommendations of the Polish Society of Allergology regarding the qualification of people with allergies and anaphylaxis for vaccination against COVID-19, patients with a history of severe hypersensitivity reactions to drugs, foods, or physical factors should also be referred to an allergy clinic [16].

Anaphylaxis was diagnosed in 0.027% of patients who received the Pfizer-BioNTech vaccine (BNT vaccine) and 0.023% of individuals vaccinated by the Moderna vaccine (M vaccine) [17].

PEG is an excipient contained in the mRNA vaccines, as well as in multiple drugs and cosmetic products. Although PEG is generally considered safe, it can cause mild to life-threatening immediate-type hypersensitivity [18].

In our study, 21 patients underwent the BAT with the vaccine and PEG, two of whom had positive results and contraindications for vaccination with PEG-containing vaccines. By confirming that PEG was the vaccine ingredient responsible for anaphylaxis, patients allergic to it could be vaccinated with a vaccine based on ingredients other than PEG. The remaining patients were vaccinated with the PEG-containing vaccine without complications.

Due to its widespread distribution, the precise diagnosis of a PEG allergy is crucial. As Sellaturay et al. point out, allergy STs, even SPTs with PEG, can trigger severe allergic reactions. The patients described by the authors developed an anaphylaxis reaction 2 min after positive results from an ST [19].

Observations by American authors indicate the limited usefulness of STs with PEG in diagnosing hypersensitivity to mRNA vaccines. Of patients with suspected allergic reactions to mRNA COVID-19 vaccines who underwent STs and BATs, 0 out of 11 tested positive to PEG in STs, while 10 out of 11 (91%) had positive BAT results to PEG and one out of 10 (10%) tested positive to the same brand of mRNA vaccine in the STs, and 11 out of 11 (100%) had positive BAT results to their administered mRNA vaccine. No PEG IgE was detected [20].

A positive BAT to PEG has been reported in small series and case reports. In clearly diagnosed PEG-allergic patients, maximal CD63% activation with the Comirnaty (Pfizer Inc., New York, NY, USA) vaccine in one study with three patients was 51%, 64.2%, and 82.1% [21,22].

In another study with the BNT vaccine, the results in patients who were supposed to be allergic were expressed in stimulation indices (SI), with values of 2.88, 3.1, 3.19, and 4.79 [23].

In our study, in the first PEG-allergic patient, we obtained positive results with the Comirnaty (Pfizer Inc. New York, USA) vaccine: 12.9% (SI value: 25.8). In the second PEG-allergic patient, we obtained negative results with the vaccine: 4.8% (SI value: 48).

In a study conducted by Warren et al., 12 out of 13 patients, who were suspected of being allergic to the mRNA vaccine, had DMG-PEG 2000-induced values from 10% to 73% [24].

In a study conducted by Labella et al., PEG 2000 SIs of 3.1. and 4.57 were found in two patients with suspected BNT vaccine allergies [23]. In a study conducted by Restivo et al. on PEG 4000, maximal CD63% activations were 14.79% and 16.2% [25].

We received positive results in our patients with DMG-PEG 2000: 11.2% and 12.2 % of activated basophils (SI: 22.4, 122) and with PEG 4000: 9.6% and 14.7% (SI 19.2, 147).

It has been reported that the BAT may be useful for assessing COVID-19 vaccines; however, according to Labella M. et al.’s study, a positive BAT result with the vaccine could indicate a past COVID-19 infection instead of an allergy [23].

There is no doubt that the BAT can be used as a potential diagnostic tool for confirming and excluding allergies to PEG excipients. In our study, the sensitivity and specificity of BAT were, respectively, 40% and 75% for the vaccine, 67% and 58% for DMG-PEG 2000, and 100% and 75% for PEG 4000.

In our study, 21 patients underwent the BAT with the vaccine and PEG, two of whom had a positive reaction and contraindications for vaccination with PEG-containing vaccines. By confirming that PEG was the vaccine ingredient responsible for anaphylaxis, patients could then be vaccinated with a vaccine based on ingredients other than PEG. The remaining patients were challenged with the PEG-containing vaccine without complications. Moreover, the clinical implications of BAT results include avoiding PEG-containing medications and preparations.

The limitations of the presented paper include small group sizes and a lack of consent from patients to perform skin tests and controlled provocation challenge tests in hospital conditions. Due to the low number of patients, further studies should be performed in a multicenter setting and should include provocation tests as a gold standard.

There are also some limitations to the BAT that must not be forgotten. Approximately 10% of basophils do not respond to stimulation through FcεRI and do not upregulate CD63 effectively due to an IgE-independent stimulus [26]. Various factors can affect the results of the BAT, for instance, the time between blood collection and the performance of the BAT, medication that the patient being tested may be on, the material used for basophil stimulation, the antibodies used for the staining of key markers, and flow cytometry analyses [26]. Blood basophils are best used fresh, ideally on the same day or up to 24 h after blood collection [27]. Individuals being tested with the BAT should stop treatment with oral steroids 3 weeks before the test, while antihistamines do not influence the result of the BAT [8].

### 4.4. The Usefulness of BAT in the Diagnosis of DIA after Other Drugs

According to data from the European Anaphylaxis Registry, the most common factors causing DIA are analgesics (41.27%), antibiotics (33.17%), local anesthetics (7.38%), radiocontrast media (5.18%), antineoplastic and immunomodulating agents (3.64%), and other drugs, mainly proton pump inhibitors (2.70%) and other drugs (6.67%) [11].

In our study, 25 patients had a history of anaphylaxis after antibiotics, and nine patients had positive BAT results. STs for penicillin have a high negative predictive value (93%); however, the sensitivity and the positive predictive value are lower [28]. Moreover, for the evaluation of a penicillin allergy, STs require major and minor determinants because patients with anaphylaxis are likely to be sensitized to minor determinants, which are not currently commercially available [7]. Skin testing for other antibiotics has not been well-validated [7].

Serum sIgE testing for antibiotic allergies is not recommended because of low sensitivity and specificity [29]. Some patients may also be sensitized (i.e., positive ST result) but not clinically allergic, with a negative oral challenge [30].

The advantages of the BAT are the ability to work with both conjugated and free drugs and rapidly incorporate new antigenic determinants. The BAT was positive in 14.3% of cases with negative STs and sIgE results with beta-lactam antibiotics [31]. In the study by Thinnes et al., the BAT was positive in nine of 12 cases with a positive clinical history but negative ST results. Furthermore, all five patients who reported severe drug hypersensitivity reactions had positive BATs. At the same time, only three of these five cases showed a positive ST [32].

In a multicenter study performed by the European Network in Drug Allergy, results were analyzed from 181 patients with a history of immediate-type allergic reactions to beta-lactams and 81 controls. In this study, BAT sensitivity was 50%, and specificity ranged from 89% to 97% [33]. A study of 18 patients with cefazolin perioperative anaphylaxis confirmed by IDT found that BATs using CD63 or CD203c were 38–75% sensitive, respectively [34].

In amoxicillin-allergic patients, activation markers of basophils, CD63 and CD203c, showed sensitivity values of 48.6% and 46.7% with very good specificities of 81.1% and 94.6% [35].

In our study, in the conditions of everyday clinical practice, the sensitivity and specificity of BAT were, respectively, 100% and 100% for amoxicillin, 60% and 86% for amoxicillin with clavulanic acid, 80% and 67% for cefuroxime, 100% and 100% for ampicillin, and 100% and 55% for clarithromycin.

Quick and reliable diagnoses of antibiotic allergies, especially a beta-lactam allergy, are crucial. The label of “allergy to beta-lactams” leads to the prescription of alternative, usually broad-spectrum antibiotics [36]. Placing a BAT as a first step in the diagnostic procedure can help reduce the need to perform a complete allergology work-up in almost half of patients and lower the risk of provoking allergic reactions [35].

Non-steroidal anti-inflammatory drugs (NSAIDs) are known to frequently cause drug hypersensitivity reactions, including severe symptoms, usually through non-IgE-dependent mechanisms [37]. In our study, we obtained positive results in two of 14 tested patients with NSAIDs. Due to the small group size, we did not calculate the sensitivity and specificity of the BAT in this group.

The usefulness of the BAT in the diagnosis of NSAID hypersensitivity reactions is controversial. In different authors’ studies, sensitivities varied from 0% to 80% and specificities from 40% to 100% [38].

In Marraccini P. et al.’s study, the authors evaluated 204 patients reporting drug hypersensitivity reactions, mainly caused by antibiotics (49%) and non-steroid anti-inflammatory drugs (NSAID) (37%). Patients with a discrepancy between anamnesis and the BAT underwent a challenge test with 100% specificity in the case of a challenge with NSAIDs [38].

According to data from the literature, the BAT is the only in vitro method that has been applied for the evaluation of both IgE-mediated reactions and NSAID hypersensitivity reactions. However, further studies are needed [38].

Our study obtained positive results in three of seven tested patients with local anesthetics. In an allergy clinic in China, of the 68 patients diagnosed over a 10-year period from 2009 to 2019 with suspected drug hypersensitivity reactions to local anesthetics, only six patients had positive results in skin tests and/or the BAT. The authors conclude that skin tests and the BAT may be useful in distinguishing true allergies from local anesthetics [39].

### 4.5. Patients with No Indications for a Drug Allergy Diagnosis

In our group, 60 out of 150 people referred to the allergology department had no indications for the diagnosis of drug allergies. In the experience of scientists from Katowice in Poland, out of approximately 200 patients who were refused vaccination due to a history of allergies, 85 patients could be vaccinated without any preceding diagnostic tests. As the authors emphasize, it seems that too often, patients with a history of allergy are excluded from vaccination due to a lack of knowledge of the recommendations or concerns of doctors qualifying for vaccination, which contributes to an increase in the level of patient anxiety and complete resignation from vaccination [12]. It must be assumed that most of the reported adverse reactions after vaccination were not anaphylactic but vasovagal events or signs related to anxiety.

## 5. Conclusions

The confirmation of an allergy is important for several reasons. It improves the safety profile of the diagnostic work-up, as it may defer the need for an oral drug challenge, preventing potential anaphylactic reactions.

The study was conducted during the COVID-19 epidemic, which gave it a specific context. Vaccination against COVID-19 was seen as an opportunity to prevent serious health consequences and potentially even death. Therefore, it was particularly important to exclude possible life-threatening contraindications, which undoubtedly include the risk of anaphylaxis.

In terms of drug allergies, the BAT may be useful if there is no drug source to use for STs or sIgE determination; there is discordance between anamnesis and ST or sIgE determination; symptoms suggest that an ST may result in a systemic response before considering a provocation test.

Currently, the BAT is applied in research settings; however, the test can be considered as a diagnostic tool for daily practice for selected patients and selected drugs when the test is available, particularly for patients who experienced severe reactions and when a diagnosis cannot be established, in order to avoid unnecessary drug provocation tests.

## Figures and Tables

**Figure 1 diagnostics-14-02036-f001:**
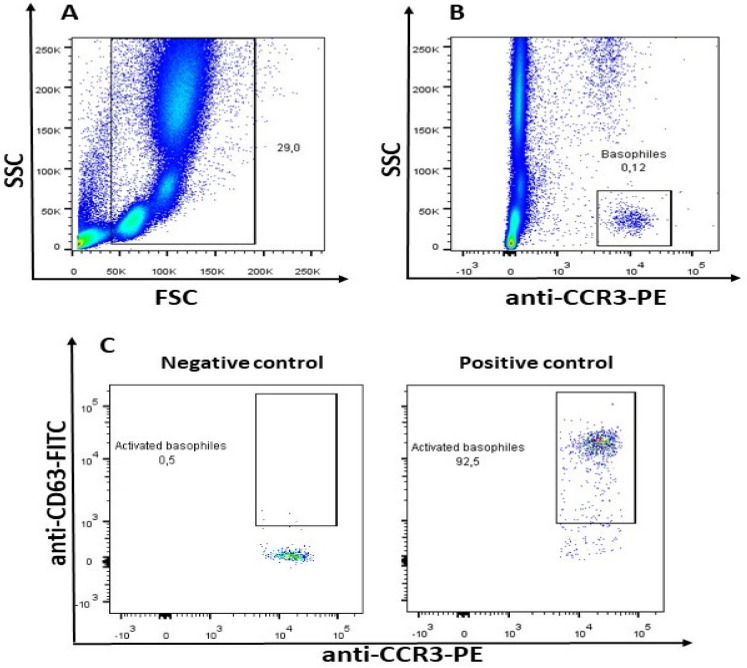
Result of Basophil Activation Test cytometric analysis; (**A**) Discrete cell populations (lymphocytes, monocytes, and granulocytes) of hemolyzed whole blood in FSC/SSC histogram; (**B**) Selection of entire basophil population on the basis of positive CCR3 and low Side Scatter (SSC); (**C**) Selection of activated fraction of basophils on the basis of high expression of CD63.

**Figure 2 diagnostics-14-02036-f002:**
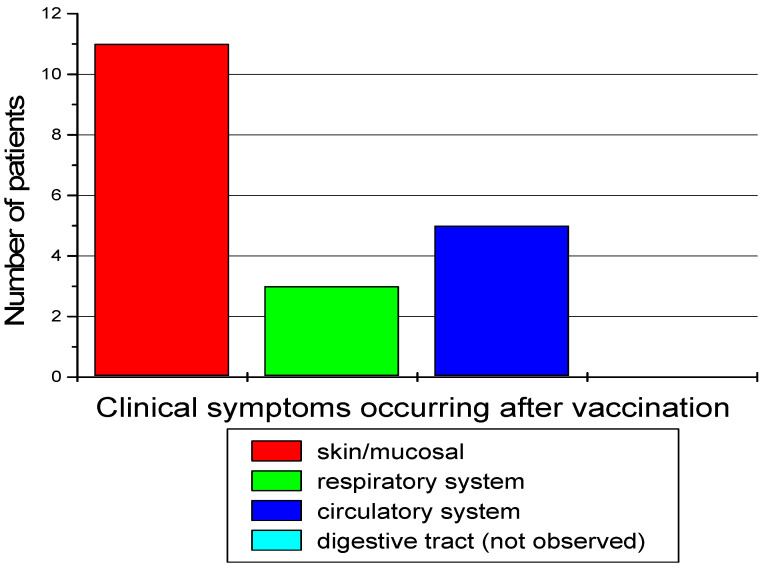
Clinical symptoms occurring after vaccination against COVID-19 in patients.

**Figure 3 diagnostics-14-02036-f003:**
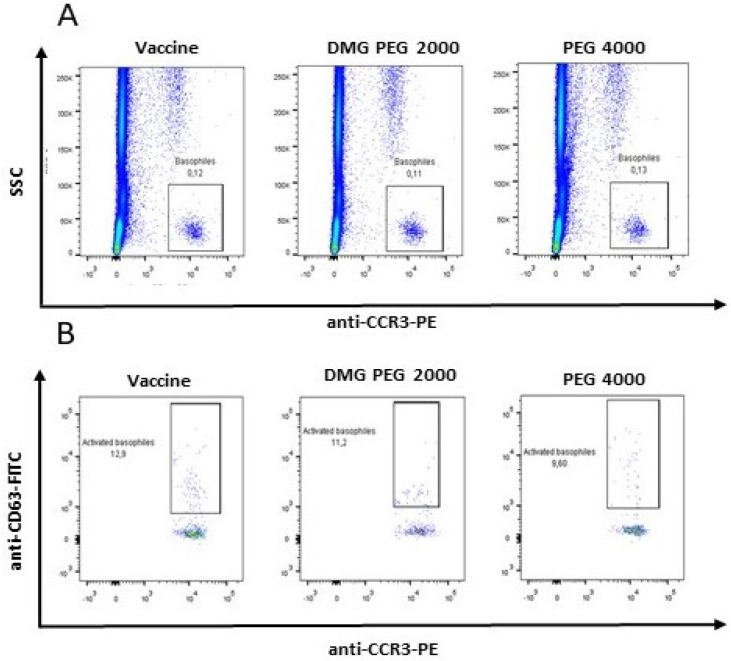
Positive BAT results with the vaccine and/or PEG for Patient 1 ((**A**) Gating strategy of basophils; (**B**) Amount of activated basophils).

**Figure 4 diagnostics-14-02036-f004:**
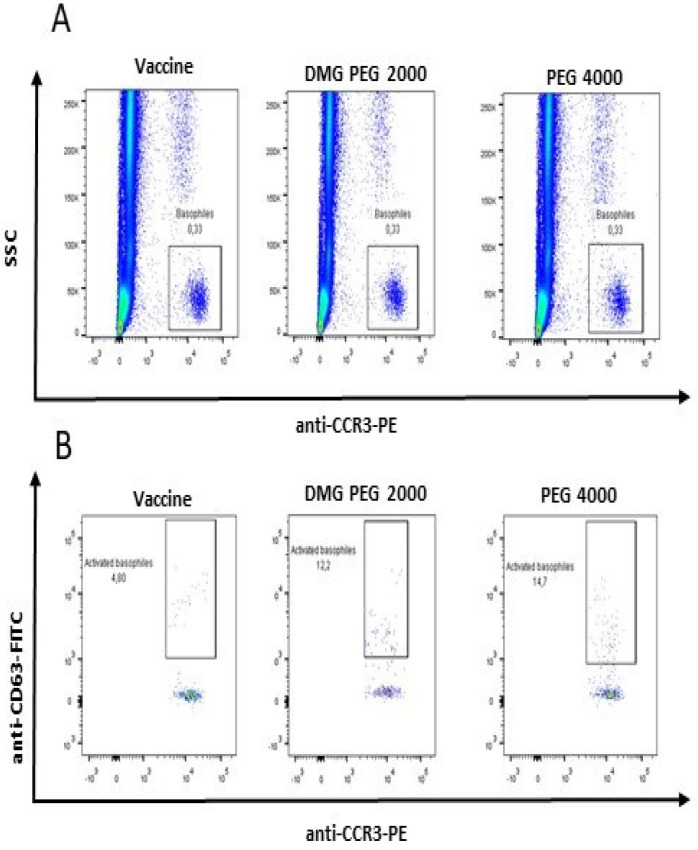
Positive BAT results with the vaccine and/or PEG for Patient 2 ((**A**) Gating strategy; (**B**) Amount of activated basophils).

**Figure 5 diagnostics-14-02036-f005:**
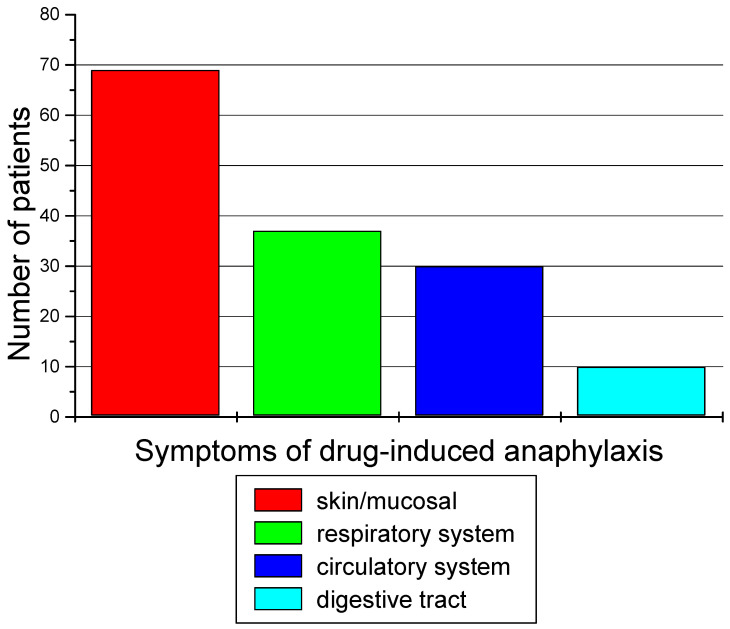
Symptoms of drug-induced anaphylaxis.

**Figure 6 diagnostics-14-02036-f006:**
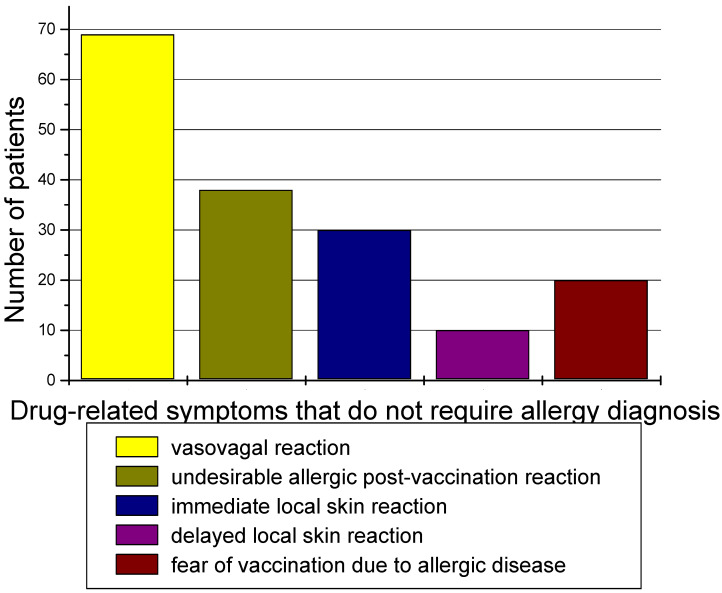
Drug-related symptoms that do not require an allergy diagnosis.

**Table 1 diagnostics-14-02036-t001:** Positive BAT and skin testing results with the vaccine and/or PEG.

No.	Age	Gender	Severity	BAT Vaccine	SPT Vaccine	IDT Vaccine	BAT PEG 2000	BATPEG 4000
1	40	F	III	+	-	+	+	+
2	24	F	III	-	-	-	+	+

+ means positive, - means negative.

**Table 2 diagnostics-14-02036-t002:** Positive BAT results.

Drug No.	Culprit Agent	Age	Gender	Severity R&M	BAT	SI	% of Activated Basophils	SPT	IDT
1	ciprofloxacin	35	M	I	+	16	38.3	nt	nt
2	penicillin	42	F	III	+	70	28.1	nt	nt
3	ampicillin	31	F	III	+	65	26.1	nt	nt
4	penicillin	49	M	III	+	5.3	11.7	nt	nt
5	cefuroxime	68	F	III	+	13.1	26.2	-	+
6	clarithromycin	46	F	II	+	22.6	11.3	nt	nt
7	amoxicillin	36	F	III	+	140.4	70.2	nt	nt
8	penicillin	68	F	III	+	56.2	28.1	nt	nt
9	ampicillin	42	F	III	+	52.2	26.1	nt	nt
10	amoxicillin + clavulanic acid	46	F	III	+	17.65	70.6	-	+
11	cefuroxime	64	F	II	+	13.1	26.2	-	+
12	amoxicillin + clavulanic acid	31	F	II	+	2.2	6.5	nt	nt
13	cefuroxime	31	F	III	+	2.1	6.2	nt	nt
14	penicillin	19	M	III	+	2	6.9	nt	nt
15	ibuprofen	23	M	II	+	3.6	13.6	nt	nt
16	acetylsalicylic acid	62	F	II	+	5.4	7.1	nt	nt
17	lidocaine	19	F	II	+	5.4	13.5	nt	nt
18	lidocaine	21	F	II	+	2.3	7.3	nt	nt
19	lidocaine	20	F	II	+	2.8	10	nt	nt

nt—not tested. + means positive, - means negative.

## Data Availability

The data presented in this study are available on request from the corresponding author. The data are not publicly available due to reasons of sensitivity.

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
