# Peer review of "The Role of the Basophil Activation Test in the Diagnosis of Drug-Induced Anaphylaxis"

_diagnostics, 2024, doi:10.3390/diagnostics14182036_

Round 1

Reviewer 1 Report

Comments and Suggestions for Authors

The article presents a study performed to assess the usefulness of BAT in the diagnosis of DIA. There are numerous areas which need to be addressed:

Introduction:

1.       Page no. 2, line no. 53-54: “…. and to avoid the discontinuation of an important treatment (e.g., antihistamine drugs) before in vivo testing, an in vitro test should be implemented”; please re-phrase the sentence for better clarity.

2.       A brief account of other available in-vitro tests such as measurement of serum mediators such as tryptase, histamine, serum specific IgE may be added along with their specific applications, sensitivity and specificity.

3.       The emerging role of mast cell activation test (MAT) may also be added.

4.       What was the research hypothesis?

5.       The aim of the study needs to be focussed in terms of study population/ indication.

Methods: the following information needs to be added:

1.       Description of study setting (primary/ secondary/tertiary care; public/ private; outpatient clinic/ in-patients; location of study site etc.).

2.       What was the study period and duration?

3.       Inclusion/ exclusion criteria of study participants.

4.       How many patients were screened for study eligibility?

5.       Were there any study withdrawals/ drop-outs?

6.       Was the questionnaire administered by the investigators/ self-administered by the patients?

7.       Was the questionnaire validated before use in the study?

8.       What was the basis of classifying patients into 3 groups from survey/ questionnaire? This needs to be explained in detail. The authors may also add a brief description of the questionnaire used per se.

9.       Were the patients asked about any vaccine-related allergy in the past as well?

10.   What was the basis of doing the study in 150 patients? Was any sample size calculation performed  a priori?

Results: Some discrepancies in the results need to be addressed:

1.       As per the text, 9 patients had positive BAT with 14 different antibiotics while these figures do not exactly correspond to the data in table 2. Please confirm.

2.       What were the BAT results in 7 patients administered radiocontrast media?

3.       How many patients were approached for IDT and how many consented?  

   Discussion

1.       The role of the BAT in the qualification for vaccination against COVID-19”…..the sentence seems incomplete.

Comments on the Quality of English Language

Moderate editing required.

Author Response

Comments and Suggestions for Authors

The article presents a study performed to assess the usefulness of BAT in the diagnosis of DIA. There are numerous areas which need to be addressed:

Thank you for all your kind comments. We have tried to include them as best as we can in the revised version of the text.

Introduction:

  1. Page no. 2, line no. 53-54: “…. and to avoid the discontinuation of an important treatment (e.g., antihistamine drugs) before in vivo testing, an in vitro test should be implemented”; please re-phrase the sentence for better clarity. - This section has been rephrased to the text.

  1. A brief account of other available in-vitro tests such as measurement of serum mediators such as tryptase, histamine, serum specific IgE may be added along with their specific applications, sensitivity and specificity. We decided not to include information about these methods as we have the impression that they do not offer sufficient specificity and do not contribute much to the diagnosis of the culprit drug in anaphylaxis.

  1. The emerging role of mast cell activation test (MAT) may also be added. This section has been added to the text.

  1. What was the research hypothesis? – This section has been added to the text.
  2. The aim of the study needs to be focussed in terms of study population/ indication. - This section has been added to the text.

Methods: the following information needs to be added:

  1. Description of study setting (primary/ secondary/tertiary care; public/ private; outpatient clinic/ in-patients; location of study site etc.). - This section has been added to the text.

  1. What was the study period and duration? - This section has been added to the text.

  1. Inclusion/ exclusion criteria of study participants. - This section has been added to the text.

  1. 4. How many patients were screened for study eligibility? - This section has been added to the text.

  1. 5. Were there any study withdrawals/ drop-outs? This section has been added to the text.

  1. 6. Was the questionnaire administered by the investigators/ self-administered by the patients? - This section has been added to the text.

  1. 7. Was the questionnaire validated before use in the study? The questionnaire was for internal use, not validated due to the pandemic regime.
  2. What was the basis of classifying patients into 3 groups from survey/ questionnaire? This needs to be explained in detail. - This section has been added to the text. The authors may also add a brief description of the questionnaire used per se. – We can add this as an Appendix

  1. Were the patients asked about any vaccine-related allergy in the past as well? Yes

  1. What was the basis of doing the study in 150 patients? Was any sample size calculation performed a priori? - This section has been added to the text.

Results: Some discrepancies in the results need to be addressed:

  1. As per the text, 9 patients had positive BAT with 14 different antibiotics while these figures do not exactly correspond to the data in table 2. Please confirm. – All the discrepancies in the table have been corrected.

  1. What were the BAT results in 7 patients administered radiocontrast media? - This section has been added to the text.
  2. How many patients were approached for IDT and how many consented- This section has been added to the text.

   Discussion

  1. “The role of the BAT in the qualification for vaccination against COVID-19”…..the sentence seems incomplete. - Corrected in the text

Reviewer 2 Report

Comments and Suggestions for Authors

Thank you for the opportunity to review this interesting manuscript. Please find my comments and remarks below.

Main issues:

1.      I suggest the Authors consider to comment on why the Ring-Messmer scale was chosen to assess anaphylactic reaction, out of many available, in particular in the context of the guidelines issued by WAO in 2020 and updated in 2023.

2.      I suggest to describe Group 3 in more detail, indicating drugs which were suspected to cause reaction that did not, whatsoever, require further diagnosis.

Minor issues:

Lines 156-157: This sentence is not quite clear and needs rephrasing

Please check that IDT abbreviation is used consistently throughout the manuscript – sometimes there is ITD used – presumably it applies to the same test, but please clarify.

Make sure that the RM abbreviation for Ring-Messmer scale is used consistently (sometimes R&M is used alternatively).

Lines 219 & 286: should be highlighted as sub-headings with different font

While mentioning Comirnaty vaccine, please indicate the manufacturer (Pfizer) and its location/headquarters site.

Comments on the Quality of English Language

Minor general English language issues, to be amnded during final editorial workup.

Author Response

Thank you for the opportunity to review this interesting manuscript. Please find my comments and remarks below.

Thank you for all your kind comments. We have tried to include them as best as we can in the revised version of the text.

Main issues:

  1. I suggest the Authors consider to comment on why the Ring-Messmer scale was chosen to assess anaphylactic reaction, out of many available, in particular in the context of the guidelines issued by WAO in 2020 and updated in 2023. - In current publications, both scales are used, we can of course convert to the WAO scale, but it will not affect the results, because only one of our patients had loss of consciousness, so it would be grade 5 according to WAO, and in our study it is grade 4 according to MRI.

  1. I suggest to describe Group 3 in more detail, indicating drugs which were suspected to cause reaction that did not, whatsoever, require further diagnosis. - This section has been added to the text.

Minor issues:

Lines 156-157: This sentence is not quite clear and needs rephrasing - This section has been rephrased in the text.

Please check that IDT abbreviation is used consistently throughout the manuscript – sometimes there is ITD used – presumably it applies to the same test, but please clarify. - This section has been corrected in the text.

Make sure that the RM abbreviation for Ring-Messmer scale is used consistently (sometimes R&M is used alternatively). - This section has been corrected in the text.

Lines 219 & 286: should be highlighted as sub-headings with different font. - This section has been corrected in the text.

While mentioning Comirnaty vaccine, please indicate the manufacturer (Pfizer) and its location/headquarters site. - This section has been corrected in the text.

Reviewer 3 Report

Comments and Suggestions for Authors

Czarnobilska and colleagues submitted an original article about the assessment of BAT and its relevance in clinical practice. The idea is highly relevant, and several allergologists are discussing such kind of advanced/molecular tests to predict or stratify patients risk score for allergies or for the detection/diagnosis of allergies. However, although the general statement on BAT is very controversial and is still investigated for academic purposes, more translational research is necessary to analyze and understand the clinical relevance of BAT. 

I have some comments, which should be addressed:

Please shorten the first half of the introduction and completey focus on the relevance of BAT. What is BAT, how does it work, what is known etc. I would rather suggest to start with this relevance instead of writing a long passage of allergies, which is not so much relevant for readers who are allergologists :)

Please provide specifically the antibodies used for FC, purchased where etc, the staining protocol, negative positive controls, FMO, incl. figures for the supplementary material

Figure 2: please use another program to create a figure and change the colors between organs. Please extent the symptoms in detail e.g., in a separate table for example 

The quality of Fig 3 ist significantly poor. Instead of a screenshot, please download the files and add them to the article. Where is the main population? Show the gating strategy!

The for Fig 4 and Fig 5

Please completely rewrite the discussion section, as this does not satisfactorily covers the content and the results of the article completely. Please focus in a later passage on the pro's and con's of BAT.

A detailed limitation section is missing.

Comments on the Quality of English Language

The English of the article is fine.

Author Response

I have some comments, which should be addressed:

Thank you for all your kind comments. We have tried to include them as best as we can in the revised version of the text.

Please shorten the first half of the introduction and completey focus on the relevance of BAT. What is BAT, how does it work, what is known etc. I would rather suggest to start with this relevance instead of writing a long passage of allergies, which is not so much relevant for readers who are allergologists :)

  • We have tried to revise this section but the reviewer 1 had opposite comments therefore we have tried to balance them accordingly :)

Please provide specifically the antibodies used for FC, purchased where etc, the staining protocol, negative positive controls, FMO, incl. figures for the supplementary material

- The information about antibodies used for flow cytometry as well as staining protocol were described in Materials and Methods section. FMO was not performed.

Figure 2: please use another program to create a figure and change the colors between organs. Please extent the symptoms in detail e.g., in a separate table for example

  • Figure 2 has been modified according to the suggestion

The quality of Fig 3 ist significantly poor. Instead of a screenshot, please download the files and add them to the article. Where is the main population? Show the gating strategy!

  • Figure 3. has been changed. The entire population of basophiles has been added for Patient 1 and 2. After modification Figure 3 presents positive BAT results for Patient 1 (A-Gating strategy; B-Amount of activated basophiles). While added Figure 4 presents results for Patient 2.

The for Fig 4 and Fig 5

  • Figure 4 and 5 have been modified.

Please completely rewrite the discussion section, as this does not satisfactorily covers the content and the results of the article completely.

  • We have made some corrections but the other reviewer praised our discussion and once more we have had to balance somewhere in between.

Please focus in a later passage on the pro's and con's of BAT.

– This section has been added to the text.

A detailed limitation section is missing.

– This section has been added to the text.

Round 2

Reviewer 1 Report

Comments and Suggestions for Authors

No further comments.

Reviewer 3 Report

Comments and Suggestions for Authors

The comments were adressed.